# Serum fetuin-A is decreased in cirrhotic patients with Wilson's disease

**Krisztián Vörös**[1]\*, **Bernadett Márkus**[1], **Klára Atzél**[1], **Ferenc Szalay**[2], **László Gráf**[3], **Dániel Németh**[2], **Tamás Masszi**[3], **Péter Torzsa**[1], **László Kalabay**[1,3]

**1** Department of Family Medicine, Semmelweis University, Budapest, Hungary, **2** Department of Internal Medicine and Oncology, Semmelweis University, Budapest, Hungary, **3** Department of Internal Medicine and Hematology, Semmelweis University, Budapest, Hungary

\* voros.krisztian@med.semmelweis-univ.hu

**Data Availability Statement:** Data cannot be shared publicly as the small case numbers could potentially lead to deductive disclosure of individuals. Data are available from the Semmelweis University Regional and Institutional

## Abstract

### Introduction

Wilson's disease may lead to cirrhosis, but timely medical treatment could slow down its progression. Clinical markers helping early diagnosis are essential. Decreased fetuin-A concentration has been reported in cirrhosis of different etiologies. The aim of this study was to investigate whether decreased serum fetuin-A concentration could identify patients with Wilson's disease who developed cirrhosis.

### Materials and methods

In this cross-sectional study we determined the serum fetuin-A concentration of 50 patients with Wilson's disease. We analyzed the data of patients with liver involvement, comparing cirrhotic and non-cirrhotic patients.

### Results

Among patients with liver involvement those with cirrhosis had significantly lower fetuin-A and albumin level, white blood cell and platelet count. Fetuin-A negatively correlated with disease duration, bilirubin level, positively with total protein and albumin concentration, but not with copper and ceruloplasmin concentrations or markers of systemic inflammation. In multivariate analysis with fetuin-A and the Nazer score or its parameters only fetuin-A was a significant determinant of having cirrhosis. In receiver operator curve analysis among patients with liver involvement the fetuin-A level of 523 μg/ml was associated with cirrhosis with 82% sensitivity and 87% specificity. The presence of the H1069Q mutation was not associated with alteration in fetuin-A concentration.

### Conclusions

The serum concentration of fetuin-A is a sensitive marker of liver cirrhosis in Wilson's disease, independently of the H1069Q mutation, ceruloplasmin concentration or systemic inflammation.

Committee of Science and Research Ethics, for researchers who meet the criteria for access to confidential data. Contact information: Semmelweis University Regional and Institutional Committee of Science and Research Ethics, e-mail: titkarsag.kutatasetikai-bizottsag@semmelweis-univ.hu.

**Funding:** The procurement of some laboratory supplies was supported by the grant of the Hungarian Ministry of Health (ETT 278/2003). The authors received no financial support. The funder had no role in study design, data collection and analysis, decision to publish, or preparation of the manuscript.

**Competing interests:** The authors have declared that no competing interests exist.

## Introduction

Wilson's disease is a rare inherited autosomal recessive disorder of copper transport with hepatic involvement leading to progressive impairment of liver function over time [1]. This process could terminate in cirrhosis requiring liver transplantation. The Nazer score has been used to grade the severity and prognosis of Wilson's disease, by assessing serum bilirubin and aspartate aminotransferase (AST) elevation, and prothrombin time prolongation [2]. It is also used to aid decision on liver transplantation [3]. Liver damage and transplantation could be prevented with proper and early treatment of Wilson's disease [4], thus clinical markers that help the early detection of cirrhosis are essential.

Human fetuin-A (also known as α2HS-glycoprotein) is a glycoprotein involved in a wide range of biological processes [5,6]. In adults, fetuin-A is almost exclusively produced by hepatocytes [7], making it a promising molecule to evaluate liver function. Decreased serum fetuin-A concentration has been reported in patients with cirrhosis of different etiologies [8,9].

Decreased serum ceruloplasmin level is characteristic in Wilson's disease. The genes of ceruloplasmin and fetuin-A along with those of transferrin and pseudocholinesterase are mapped to the 3q21-3qter region of the 3rd chromosome [10] thus their regulation could be related. The H1069Q mutation is present in 15–72% of European patients with Wilson's disease [11] and it is the most common mutation in Hungary, as well [12]. It has been linked to predominantly neurological manifestations [13], while other, non-H1069Q mutations have been associated with early severe liver disease [14].

The aim of this study was to investigate whether decreased serum fetuin-A concentration could be a helpful parameter to identify patients with Wilson's disease who developed cirrhosis, and whether genetic alterations and ceruloplasmin levels influence serum concentration of fetuin-A and thus could hinder its diagnostic use.

## Materials and methods

### Patients

Fifty patients with Wilson's disease (29 men, 21 women, age: 33.6 ± 12.4 years, mean ± SD, duration of disease: 11.4 ± 7.1 years) were involved in the study. Diagnosis of Wilson's disease was based on the Leipzig score [15] and only data of patients with a Leipzig score ≥ 4 were included. Liver involvement was diagnosed if any of the following were detected: aspartate aminotransferase > 50 U/l, prothrombin time > 19 s, or serum bilirubin > 34 μM/l. The diagnosis of liver cirrhosis was based on hepatosplenomegaly, abdominal ultrasound, Fibroscan and histological findings. Exclusion criteria were as follows: liver involvement of viral (hepatitis B virus surface antigen and anti-HCV positivity) or autoimmune etiology (antinuclear, anti-smooth muscle, anti-mitochondrial antibody positivity), liver cancer, and treatment with hepatotoxic drugs. No liver-transplanted patients were included. Patients were on penicillamine or trientine therapy. The study was approved by the ethics committee of Semmelweis University (Semmelweis University Regional and Institutional Committee of Science and Research Ethics, 62/2019) and complied with the guidelines of the Declaration of Helsinki. All patients gave written informed consent.

### Determination of serum fetuin-A concentration

Serum levels of fetuin-A were determined by radial immunodiffusion using 10x10 cm slides as described previously [16]. In brief, 5 μl of patient's sera diluted to 1:4 was applied in 11.5 ml of Litex agarose gel (Sigma). Serum samples (1:4 dilution) with known concentrations of fetuin-A served as standards. The incubation was performed at room temperature for 48 hours. We

used anti-fetuin-A (IgG fraction, Incstar, Cat No. 81931, 13.7 mg/ml, in a final concentration of 84 µl/11.5 ml gel) as antibody. The intraassay (IACV) and interassay (IECV) variations were 3.6% and 6.2%, respectively.

## Other laboratory determinations

The positive acute phase reactant serum 1-acid glycoprotein (orosomucoid, AGP) concentration was also determined by RID using anti-human antibodies produced in goats (DiaSorin, Stillwater, Mi, USA, IgG fraction, Cat No. 81901 and 91913). The IACV and IECV were 4.2% and 5.0%, respectively. The H1069Q mutation of the ATP7B was investigated by an restriction fragment polymorphism method based on semi-nested polymerase chain reaction [12]. The other laboratory parameters were determined by conventional standardized methods.

## Statistical analysis

Contingency tables were analyzed by the Fisher's exact test and Chi-square test. For the comparison of ordinal or continuous variables Mann-Whitney U test, for multiple groups Kruskal-Wallis tests were used. Correlation studies were done by the rank correlation analysis (Spearman). We performed multivariate regression analysis to test whether low concentration of fetuin-A is independently associated with cirrhosis after adjusting for other clinical parameters. For the detection of collinearity between predictors, we used the method described at the statistical software manufacturers site [17]. We used receiver operating characteristic (ROC) analysis to determine the optimal cutoff point of fetuin-A to differentiate between patients with or without cirrhosis. Statistical analysis was performed with the SPSS v. 25 statistical program (Armonk, NY, USA). The level of $p < 0.05$ was considered significant.

## Results

Thirty-four out of the fifty patients with Wilson's disease had liver involvement. Serum fetuin-A was not significantly different between patients with and without liver involvement (562 ± 138 vs 621 ± 154 mg/l; p = 0.126).

Among patients with liver involvement fetuin-A levels negatively correlated with the duration of the disease (r = -0.471, p = 0.005) and bilirubin levels (r = -0.489, p = 0.007) and positively with total protein (r = 0.476, p = 0.014) and albumin concentration (r = 0.382, p = 0.041), but not with INR, liver function tests, white blood cell or thrombocyte counts, copper and ceruloplasmin concentrations (all NS). We also found no correlation between fetuin-A and markers of systemic inflammation (sedimentation rate, C-reactive protein (CRP), AGP).

## Cirrhotic vs non-cirrhotic patients with Wilson's disease and liver involvement

Among the 34 patients with liver involvement, 11 suffered from cirrhosis, while 23 persons were free from cirrhosis. Patients with cirrhosis had significantly lower fetuin-A levels than those with liver involvement but no cirrhosis (486 ± 57 vs 676 ± 149 mg/l; p < 0.001) (Fig 1). The large effect size (0.657) indicated a strong difference. Cirrhotic patients also had lower albumin level, white blood cell and platelet count, although the latter did not reach statistical significance (Table 1).

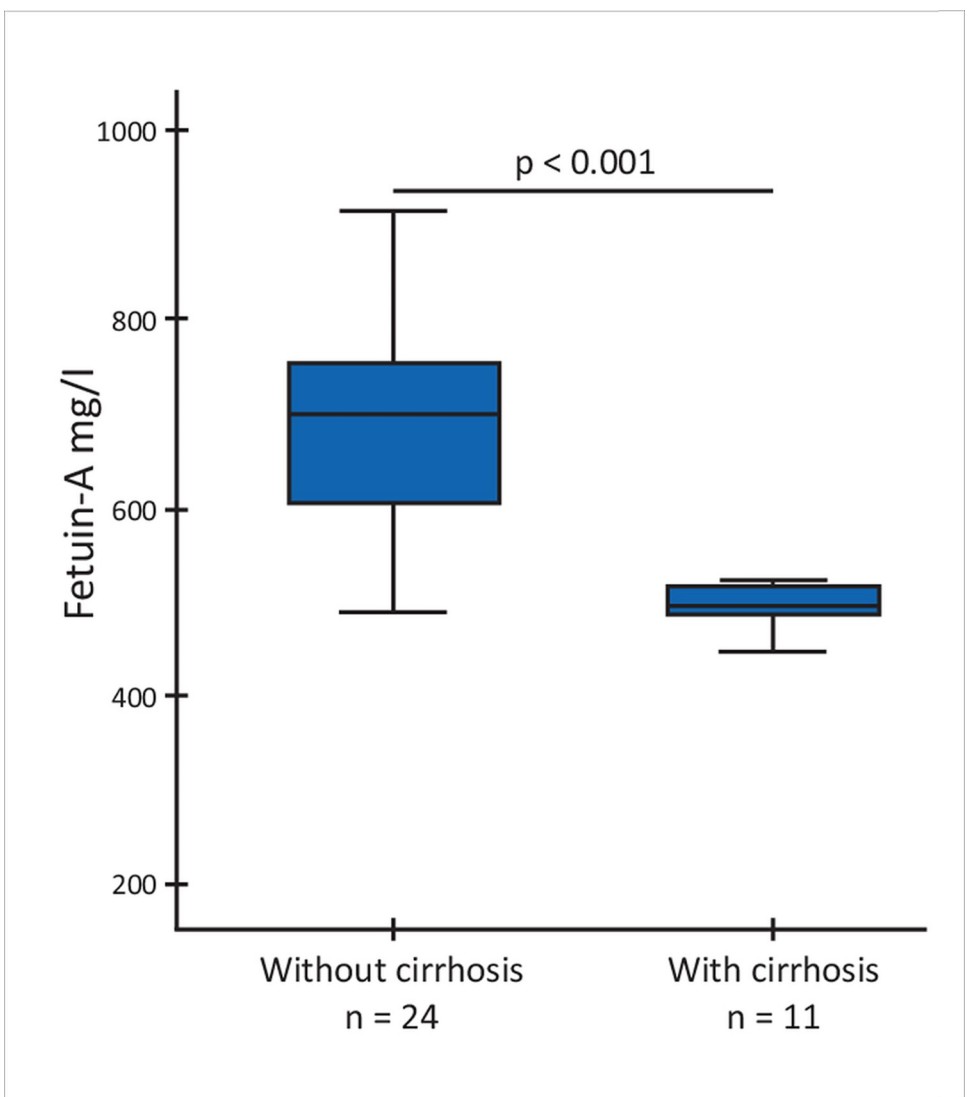

**Fig 1. Serum fetuin-A concentration in Wilson's patients with liver involvement without and with cirrhosis (Mann-Whitney U test).**

## Multivariate analysis and comparison of serum fetuin-A concentrations with the Nazer score among patients with liver involvement

Ten patients had elevated Nazer score. Among them six had cirrhosis (60%), while the occurrence of cirrhosis among patients with Nazer score = 0 was significantly lower (21%, $\chi^2$-test, p = 0.045). Patients with Nazer scores $\geq$ 1 (n = 10) had markedly lower serum fetuin-A levels than those with 0 score (n = 24) (661 ± 148 mg/ml vs 527 ± 130 mg/ml, p = 0.013).

Among patients with liver involvement, in multivariate logistic regression analysis with fetuin-A and the parameters used in the Nazer score (bilirubin, aspartate aminotransferase and prothrombin time) only fetuin-A was a significant determinant of having cirrhosis (Exp (B) = 0.973, CI = 0.948–0.999, p = 0.040). The model was statistically significant ($\chi^2(4) =$ 17.628, p < 0.001), explained 67.9% of the variance in cirrhosis occurrence (Nagelkerke $R^2$) and correctly classified 84.6% of cases. The multi-collinearity analysis yielded low Variance Inflation Factors (VIF) between 1.055 and 1.411, indicating minimal collinearity.

**Table 1. Comparison of laboratory parameters of Wilson patients with non-cirrhotic and cirrhotic liver involvement (mean ± SD).**

| Parameter | Liver involvement without cirrhosis (n = 23) | Liver cirrhosis (n = 11) | p |
|---|---|---|---|
| Age (years) | 32 ± 12 | 35 ± 10 | 0.224 |
| Age at the time of diagnosis (years) | 22 ± 11 | 23 ± 6 | 0.354 |
| Duration of disease (years) | 9 ± 5 | 13 ± 7 | 0.164 |
| Body mass index (kg/m$^2$) | 23.6 ± 5.2 | 23.5 ± 3.3 | 0.861 |
| **Fetuin-A (mg/l)** | **676 ± 149** | **486 ± 57** | **0.001** |
| C-reactive protein (mg/l) | 1.2 ± 1.1 | 0.3 ± 1.12 | 0,093 |
| Erythrocyte sedimentation rate (mm/h) | 6 ± 10 | 3 ± 2 | 0.889 |
| Red blood cell count (10$^6$/μl) | 5.0 ± 0.4 | 4.8 ± 0.5 | 0.322 |
| Hematocrit | 0.44 ± 0.04 | 0.41 ± 0.06 | 0.635 |
| Hemoglobin (g/l) | 151 ± 11 | 140 ± 26 | 0.524 |
| **White blood cell count (/μl)** | **6375 ± 1605** | **5144 ± 1269** | **0.043** |
| Platelet count (10$^3$/μl) | 202 ± 70 | 156 ± 77 | 0.114 |
| Serum creatinine (μmol/l) | 84 ± 18 | 80 ± 20 | 0.423 |
| eGFR (ml/min/1.73m$^2$) | > 60 | > 60 | 0.858 |
| INR | 1.10 ± 0.10 | 1.30 ± 0.48 | 0.186 |
| Prothrombin time (s) | 13.8 ± 1.6 | 15.8 ± 4,5 | 0.256 |
| Total cholesterol (mmol/l) | 4.5 ± 0.9 | 5.0 ± 0.7 | 0.089 |
| Triglyceride (mmol/l) | 1.1 ± 0.7 | 1.3 ± 0.5 | 0.140 |
| Serum bilirubin (μmol/l) | 13.9 ± 5.9 | 26.0 ± 21.9 | 0.063 |
| Aspartate aminotransferase (U/l) | 34 ± 12 | 35 ± 24 | 0.333 |
| Alanine transaminase (U/l) | 46 ± 22 | 49 ± 46 | 0.370 |
| Alkaline phosphatase (U/l) | 258 ± 221 | 413 ± 552 | 0.268 |
| γ-glutamyl-transferase (U/l) | 47 ± 42 | 116 ± 189 | 0.345 |
| Lactate dehydrogenase (U/l) | 330 ± 116 | 297 ± 66 | 0.563 |
| Total protein (g/l) | 77 ± 5 | 72 ± 2 | 0.052 |
| **Serum albumin (g/l)** | **49 ± 3** | **43 ± 9** | **0.027** |
| Serum copper (μmol/l) | 6.3 ± 4.7 | 7.8 ± 4.8 | 0.373 |
| Serum ceruloplasmin (g/l) | 0.09 ± 0.06 | 0.11 ± 0.05 | 0.193 |
| **Nazer score >0 (n)** | **1 (4.3%)** | **5 (45,5%)** | **0.022**[*] |

Mann-Whitney U test

[*] Fisher's exact test.

We also analyzed if the use of the Nazer score instead of its components might have a stronger predictive value. In a model using the Nazer score and fetuin-A as determinants ($\chi^2(2)$ = 17.542, p < 0.001, Nagelkerke $R^2$ = 65.1%), the former failed to be a significant predictor (p = 0.370), while fetuin-A remained significant (Exp(B) = 0.978, CI = 0.958–0.999, p = 0.039). The VIF values were 1.109 for both variables, indicating minimal collinearity.

## Determination of the cut-off serum fetuin-A value indicating cirrhosis in patients with liver involvement

We determined the cut-off serum fetuin-A value using ROC analysis in the sera of patients with liver involvement (n = 34). The area under the curve (AUC) was 0.911 ± 0.049, 95%CI: 0.815–1.000 (p < 0.001) (Fig 2). The fetuin-A concentration of 523 mg/l was associated with cirrhosis with 82.0% sensitivity and 87.0% specificity. Cirrhosis was present in 75% of patients

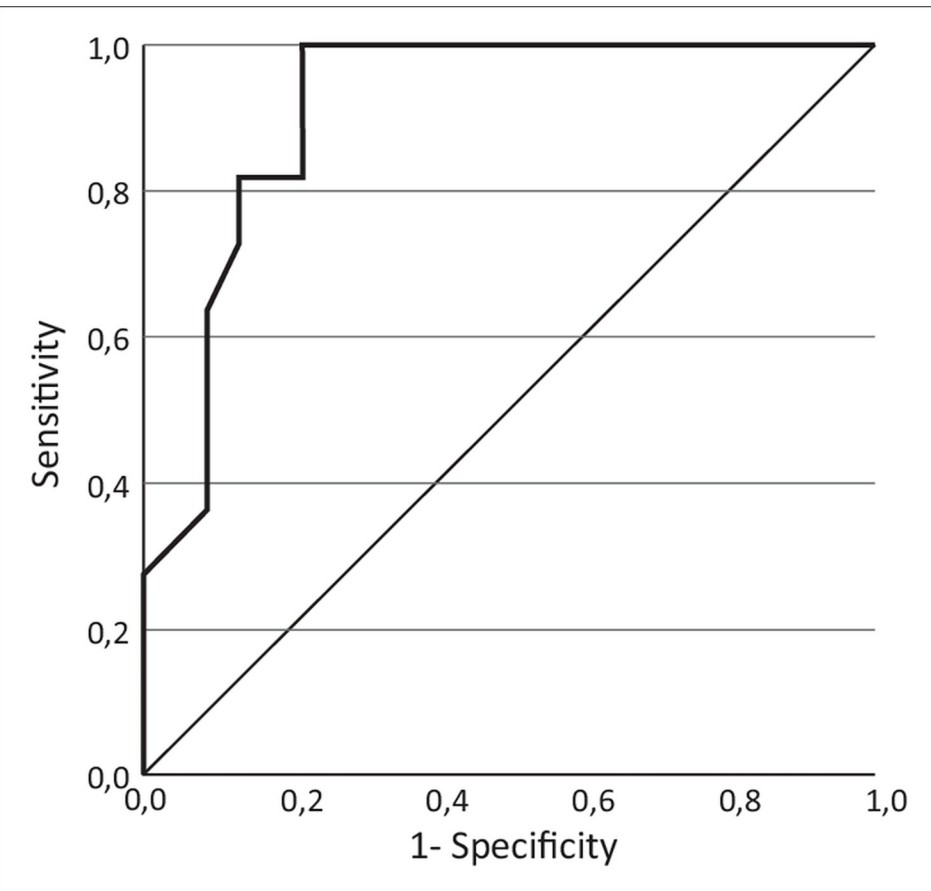

**Fig 2. Serum fetuin-A concentration receiver operating characteristic curve for having cirrhosis among patients with liver involvement (n = 34).**

with low fetuin-A concentration, while it occurred only in 9% of patients with fetuin-A concentrations above the cut-off point.

### Serum fetuin-A concentration and H1069Q mutation

H1069Q homozygous mutation was detected in 9, heterozygous in 27 patients. Serum fetuin-A levels did not differ between patients with H1069Q mutation and the wild type (Table 2).

### Discussion

Low fetuin-A concentration proved to be a helpful parameter in identifying cirrhosis in patients with Wilson's disease. Hepatic presentation ranges from asymptomatic cases to

**Table 2. Relationship between serum concentrations of fetuin-A (mean ± SD) and the H1069Q mutation.**

| | H1069Q mutation | | | |
|---|---|---|---|---|
| | **Homozygous (n = 9)** | **Heterozygous (n = 27)** | **Wild type (n = 14)** | **p*** |
| Fetuin-A mg/l | 556 ± 115 | 621 ± 161 | 597 ± 152 | 0.499 |

Kruskal-Wallis test.

cirrhosis: some degree of involvement is present is most patients with Wilson's disease at the time of the diagnosis [15], while cirrhosis may occur in up to 58% of adult patients [18]. Early recognition of cirrhosis could be challenging, as this condition often presents in a mild and insidious form: the prevalence of cirrhosis could be as high as 11% among asymptomatic patients [19]. The identification of new markers of cirrhosis could aid early detection of these mild cases. Fetuin-A may be a promising candidate for the diagnosis of cirrhosis. Although it did not differ between patients with or without liver involvement, we found a strong significant decrease of its concentration in cirrhosis, compared to non-cirrhotic patients with hepatic manifestations. In bivariate analysis fetuin-A was markedly lower, with a large effect size. In multivariate regression analysis it proved to be a stronger predictor of having cirrhosis than the Nazer score, or any of its components.

In bivariate analysis leukocyte number and serum albumin concentrations were significantly lower in cirrhotic patients, but there was no difference in transaminase concentrations. These findings are in line with a previous work, that reported smaller elevation of alanine transaminase, aspartate transaminase and bilirubin in cirrhotic patients with Wilson's disease compared to those with hepatitis B. The mean transaminase concentrations were within the normal range among patients with cirrhosis of Wilson's etiology [20]. Moreover, lack of difference in transaminase concentrations was also reported in another study, that compared cirrhotic and non-cirrhotic patients with Wilson's disease. Among the laboratory parameters investigated, Zhong et al found decreased synthetizing (lower prothrombin time) and bone marrow function (leukopenia and thrombocytopenia) but no differences for liver enzyme levels [21]. We also found proof of impaired protein synthesis, although contrary to the results of Zhong et al, the concentration of albumin was lower, while there was no significant difference in prothrombin time. In line with their results, cirrhotic patients in our study had lower white blood cell and thrombocyte counts, although the latter did not reach statistical significance. Lower thrombocyte count can be present in cirrhosis of any etiology, due to the suppression of the bone marrow by toxic substances, or hypersplenism, or by the reduced production of thrombopoetin in the liver [22]. Moreover, agents used in the treatment of Wilson's disease also cause bone marrow toxicity [23], thus the clinical utility of thrombocytopenia, leukopenia may be questioned.

Missing or weak differences in liver enzyme and liver synthetized protein concentrations and the questionable utility of cytopenia further emphasize the need for reliable parameters to detect early cirrhosis in Wilson's disease. Fetuin-A seems a promising molecule, as it was markedly lower in cirrhotic patients, independently of other laboratory parameters. It serves as a good marker of hepatic protein synthetizing capability as it is almost exclusively secreted by the liver in adults [7]. Decreased concentrations have been reported in cirrhosis of different etiologies. In a small cross-sectional study, patients with alcoholic cirrhosis had lower fetuin-A concentration compared to healthy controls [8]. In two prospective cohort studies enrolling patients with alcoholic liver cirrhosis, low concentration of the glycoprotein was an excellent indicator of short-term (one month) and long-term (1 year) survival [16,24]. Its predictive value for long-term mortality was superior to that of the Child-Pugh and Model for End-Stage Liver Disease (MELD) scores [24]. In a cross-sectional study, low concentration of fetuin-A was also reported in primer biliary cholangitis (cirrhosis) [9].

Severe forms of cirrhosis are often accompanied by low grade systemic inflammation [25] and fetuin-A is a negative acute phase reactant [26]. This association may also lead to its decreased concentration. In line with our previous studies on patients with alcoholic cirrhosis [16,24], the lack of the correlation with alpha1-acid glycoprotein and CRP indicates that the decrease of fetuin-A concentration can be attributed rather to liver parenchymal cell dysfunction than to the acute phase reaction. This is reflected by the consequent positive correlation

between fetuin-A and serum albumin in alcoholic cirrhosis, primary biliary cholangitis (cirrhosis) and in Wilson's disease, in the current study [9,16,24].

Late recognition of cirrhosis might have dire consequences. The transition to decompensated cirrhosis may happen at a 10.6% / year rate, and the median survival time in these cases is estimated to be around 2 years [27]. Symptom occurrence and severity often lead to very poor health-related quality of life in end-stage liver disease [28]. Transplantation in Wilson's disease is generally performed in acute liver failure, but end-stage liver disease is also a common indication in patients in their thirties-forties [3]. Medical treatment of Wilson patients in the decompensated phase of cirrhosis could lead to recovered liver function, further emphasizing the importance of early recognition [4].

We assessed our data to determine a cut-off point for a fetuin-A concentration identifying cirrhosis. Previously, the cut-off level of 365 mg/l was reported to be indicative for mortality due to liver failure within one year in alcoholic cirrhosis patients [24]. We found higher cut-off level in Wilson patients (523 mg/l). This difference can be explained by the fact that different outcomes were evaluated: lethal outcome in alcoholic cirrhosis and the development of cirrhosis in Wilson's disease. Values predicting mortality due to liver failure are probably much lower in Wilson's disease, as well. In addition, compared to alcoholic cirrhosis, patients with Wilson's disease were under a more careful follow-up and were treated early to prevent cirrhosis. Their cirrhosis, even if developed finally, was not as severe as in alcoholics, i.e., they showed no clinical or laboratory signs of liver failure. Although systematic follow-up was not performed, all Wilson patients survived for at least one year, while the 1-year mortality among patients with alcoholic cirrhosis was 41% [24].

We assessed whether mutation of the ATP7B gene or ceruloplasmin concentration might alter fetuin-A levels, and thus decreasing its diagnostic value. The H1069Q mutation is by far the most frequent in Europe. In our sample 36 patients had the H1069Q mutation (72.0%), which is in line with a previous study, that reported a prevalence of 64.3%, all other point mutations being uncommon [12]. Serum fetuin-A levels did not differ between patients with or without this mutation. The genes of ceruloplasmin and fetuin-A are both mapped to the 3q21-3qter region of the 3rd chromosome [10]. We found no relationship between the serum concentrations of these two molecules.

There are some limitations of this study. As Wilson's disease is a rare disorder, we managed to enroll a relatively small number of patients from one centrum. The sample size and the cross-sectional design of our study might limit its generalizability. The cirrhosis detected in patients were generally mild, however, more severe cases may even have lower fetuin-A concentration, as seen in patients with alcoholic cirrhosis [24], and thus our result may even underestimate the decrease in Wilson's disease.

## Conclusions

Our results indicate that the serum concentration of fetuin-A is a sensitive marker of liver cirrhosis in Wilson's disease, independently of the H1069Q mutation, ceruloplasmin concentration or systemic inflammation. Larger studies, involving more advanced cases are needed to evaluate whether serum fetuin-A may also bear a predictive value and have clinical importance.

## Acknowledgments

The authors thank Mrs. V.M. Nagy for her skillful laboratory assistance.

## Author Contributions

**Conceptualization:** László Gráf, László Kalabay.

**Data curation:** Krisztián Vörös, Bernadett Márkus, Klára Atzél, Dániel Németh.

**Formal analysis:** Krisztián Vörös, Bernadett Márkus, László Kalabay.

**Investigation:** Klára Atzél, Ferenc Szalay, László Gráf, Dániel Németh.

**Writing – original draft:** Krisztián Vörös.

**Writing – review & editing:** Ferenc Szalay, Tamás Masszi, Péter Torzsa, László Kalabay.

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
