## [Decision Letter · Decision Letter 0]

19 Jan 2023

PONE-D-22-32810Serum fetuin-A is decreased in cirrhotic patients with Wilson’s diseasePLOS ONE

Dear Dr. Vörös,

Thank you for submitting your manuscript to PLOS ONE. After careful consideration, we feel that it has merit but does not fully meet PLOS ONE’s publication criteria as it currently stands. Therefore, we invite you to submit a revised version of the manuscript that addresses the points raised during the review process.

We look forward to receiving your revised manuscript.

Kind regards,

Antonio De Vincentis

Academic Editor

PLOS ONE

Journal Requirements:

"The authors thank Mrs. V.M. Nagy for her skillful laboratory assistance. Start of this work was supported by the grant of the Hungarian Ministry of Health ETT 278/2003."

Reviewers' comments:

Reviewer's Responses to Questions

**Comments to the Author**

1. Is the manuscript technically sound, and do the data support the conclusions?

Reviewer #1: Yes

2. Has the statistical analysis been performed appropriately and rigorously? 

Reviewer #1: Yes

3. Have the authors made all data underlying the findings in their manuscript fully available?

Reviewer #1: Yes

4. Is the manuscript presented in an intelligible fashion and written in standard English?

Reviewer #1: Yes

5. Review Comments to the Author

Reviewer #1: This is an interesting manuscript. I have minor comments:

1: please described how would you suggest to use serum fetuin A in routine clinical care?

2: is there also predictive role of moderate fetuine A increase in the development of cirrhosis?

6. PLOS authors have the option to publish the peer review history of their article (what does this mean?). If published, this will include your full peer review and any attached files.

Reviewer #1: No

---

## [Author Response · Author response to Decision Letter 0]

7 Feb 2023

Response to reviewers

We thank the Editor and the Reviewers for having addressed several important points, which certainly help improving the quality of our manuscript. We have revised our manuscript according to all of them.

1, We have corrected the authors’ affiliations on the Title page. We have harmonized the headings of the Abstract and the Manuscript.

2, We have updated the financial statement.

The authors did not receive any funding, however at the start of the study, some reagents needed to determine fetuin-A concentrations were previously obtained by the financial help of the grant of the Hungarian Ministry of Health ETT 278/2003.

3, We have corrected the Acknowledgement section.

4, We have updated the Data Availability statement.

Small case numbers, and lack of mentioning the uploading of data to a public repository in the ethics statement makes it impossible to upload our data to a public repository. We have transferred the data to Semmelweis University Regional and Institutional Committee of Science and Research Ethics, and provided their central administrative email address as contact info (titkarsag.kutatasetikai-bizottsag@semmelweis-univ.hu translates as secretariat.researchethics-committee@semmelweis-univ.hu).

5, We have included the full ethics statement in the Methods section.

6, We did not retract citations. The ending page numbers of reference 26 and 27 broke in a new line (… 1118-29 and 217-31), resulting in empty lines starting with 29 and 31. I have replaced the normal hyphen to a non-breaking hyphen in reference 26 and 27 to solve this issue.

1: please described how would you suggest to use serum fetuin A in routine clinical care?

Fetuin-A is implicated in a wide range of diseases including type 2 diabetes, atherosclerosis, calcification of arteries and heart valves, sepsis, etc.. Although it is available in some laboratories as a liver function marker, its use in routine care is not widespread.

In theory it could be a clinical marker of insulin resistance, impaired metabolism, diabetes, as elevated concentration of fetuin-A is associated with insulin resistance, steatosis [1], and even incident diabetes [2]. However, measurement of insulin resistance, diagnosis and follow-up diabetes seem well established, where further advantages from determining fetuin-A concentration seem dubious to me.

Probably due to its deleterious effects on metabolism, elevated fetuin-A concentration was associated with atherosclerotic diseases [3, 4]. However, by decreasing calcification and inflammation it may play a protective role as well. Invers correlation was also reported in a large observational study [5], while others found no correlation [6]. I do not see any possible clinical benefit in this respect.

Fetuin-A is produced in the liver in adults, thus its use as a liver function test seems reasonable. Decreased levels were reported in severe forms of liver disease, mainly cirrhosis [Ref 8, 9, 16, 24 of the manuscript and this paper]. If larger studies will support these promising results and its role as a predictive factor will also be underpinned, its clinical use will probably be justified.

2: is there also predictive role of moderate fetuine A increase in the development of cirrhosis?

Elevated fetuin-A level is known to be associated with fatty liver disease [1], and steatosis could progress into steatohepatitis and finally cirrhosis. To our best knowledge the role of fetuin-A in this process has not been investigated.

Decreased concentrations were reported in cirrhosis in primary biliary cirrhosis (primary biliary cholangitis) and alcoholic liver disease [Ref 8, 9 of the manuscript]. However, these studies were cross-sectional. In a prospective study [Ref 24 of the manuscript] decreased levels were excellent predictors of one-year mortality, but the development or progression of cirrhosis was not investigated. Whether deceased concentrations could play a predictive role in the development, progression of cirrhosis or could aid selecting patients for transplantation warrants further studies.

References

N. Stefan, A.M. Hennige, H. Staiger, J. Machann, F. Schick, S.M. Krober, et al., Alpha2-Heremans-Schmid glycoprotein/fetuin-A is associated with insulin resistance and fat accumulation in the liver in humans, Diabetes Care. 2006; 29: 853–857.

J.H. Ix, M.L. Biggs, K.J. Mukamal, J.R. Kizer, S.J. Zieman, D.S. Siscovick, et al., Association of fetuin-A with incident diabetes mellitus in community-living older adults: the cardiovascular health study, Circulation. 2012; 125: 2316–2322.

Sommer P, Schreinlechner M, Noflatscher M, et al. High baseline fetuin-A levels are associated with lower atherosclerotic plaque progression as measured by 3D ultrasound. Atheroscler Plus. 2021; 45:10-17.

K. Vörös, L. Graf Jr., Z. Prohaszka, L. Graf, P. Szenthe, E. Kaszas, et al., Serum fetuin-A in metabolic and inflammatory pathways in patients with myocardial infarction, Eur J Clin Invest. 2011; 41: 703-709.

Ix JH, Barrett-Connor E, Wassel CL, et al. The associations of fetuin-A with subclinical cardiovascular disease in community-dwelling persons: the Rancho Bernardo Study. J Am Coll Cardiol. 2011; 58(23): 2372-2379.

Aroner SA, St-Jules DE, Mukamal KJ, et al. Fetuin-A, glycemic status, and risk of cardiovascular disease: The Multi-Ethnic Study of Atherosclerosis. Atherosclerosis. 2016; 248: 224-229.

---

## [Editor Report · Decision Letter 1]

23 Feb 2023

Serum fetuin-A is decreased in cirrhotic patients with Wilson’s disease

PONE-D-22-32810R1

Dear Dr. Vörös,

We’re pleased to inform you that your manuscript has been judged scientifically suitable for publication and will be formally accepted for publication once it meets all outstanding technical requirements.

Kind regards,

Antonio De Vincentis

Academic Editor

PLOS ONE
---

## [Editor Report · Acceptance letter]

27 Feb 2023

PONE-D-22-32810R1 

Serum fetuin-A is decreased in cirrhotic patients with Wilson’s disease 

Dear Dr. Vörös:

I'm pleased to inform you that your manuscript has been deemed suitable for publication in PLOS ONE. Congratulations! Your manuscript is now with our production department. 

Kind regards, 

on behalf of

Dr. Antonio De Vincentis 

Academic Editor

PLOS ONE